# DIFFERENCE-SEEKING GENERATIVE ADVERSARIAL NETWORK–UNSEEN SAMPLE GENERATION

**Yi-Lin Sung**
Graduate Institute of Communication Engineering
National Taiwan University, Taiwan, ROC
Institute of Information Science, Academia Sinica
r06942076@ntu.edu.tw

**Sung-Hsien Hsieh**
Institute of Information Science and
Research Center for Information Technology
Innovation, Academia Sinica, Taiwan, ROC
parvaty316@hotmail.com

**Soo-Chang Pei**
Graduate Institute of Communication Engineering
National Taiwan University, Taiwan, ROC
peisc@ntu.edu.tw

**Chun-Shien Lu**
Institute of Information Science and
Research Center for Information Technology
Innovation, Academia Sinica, Taiwan, ROC
lcs@iis.sinica.edu.tw

## ABSTRACT

Unseen data, which are not samples from the distribution of training data and are difficult to collect, have exhibited importance in numerous applications, (*e.g.,* novelty detection, semi-supervised learning, and adversarial training). In this paper, we introduce a general framework called **d**ifference-**s**eeking **g**enerative **a**dversarial **n**etwork (DSGAN), to generate various types of unseen data. Its novelty is the consideration of the probability density of the unseen data distribution as the difference between two distributions $p_{\bar{d}}$ and $p_d$ whose samples are relatively easy to collect. The DSGAN can learn the target distribution, $p_t$, (or the unseen data distribution) from only the samples from the two distributions, $p_d$ and $p_{\bar{d}}$. In our scenario, $p_d$ is the distribution of the seen data, and $p_{\bar{d}}$ can be obtained from $p_d$ via simple operations, so that we only need the samples of $p_d$ during the training. Two key applications, semi-supervised learning and novelty detection, are taken as case studies to illustrate that the DSGAN enables the production of various unseen data. We also provide theoretical analyses about the convergence of the DSGAN.

## 1 INTRODUCTION

Unseen data[1] are not samples from the distribution of the training data and are difficult to collect. It has been demonstrated that unseen samples can be applied to several applications. Dai et al. (2017) proposed how to create complement data, and theoretically showed that complement data, considered as unseen data, could improve semi-supervised learning. In novelty detection, Yu et al. (2017) proposed a method to generate unseen data and used them to train an anomaly detector. Another related area is adversarial training Goodfellow et al. (2015), where classifiers are trained to resist adversarial examples, which are unseen during the training phase. However, the aforementioned methods only focus on producing specific types of unseen data, instead of enabling the generation of general types of unseen data.

In this paper, we propose a general framework called **d**ifference-**s**eeking **g**enerative **a**dversarial **n**etwork (DSGAN), to generate a variety of unseen data. The DSGAN is a generative approach. Traditionally, generative approaches, which are usually conducted in an unsupervised learning manner, are developed for learning the data distribution from its samples, from which subsequently, they produce novel and high-dimensional samples, such as the synthesized image Saito et al. (2018). A state-of-the-art approach is the so-called generative adversarial network (GAN) Goodfellow et al. (2014). GAN produces sharp images based on a game-theoretic framework, but it can be difficult and unstable to train owing to multiple interaction losses. Specifically, GAN consists of two functions: generator and discriminator. Both functions are represented as parameterized neural networks. The discriminator network is trained to determine whether the inputs belong to the real dataset or fake dataset created by the generator. The generator learns to map a sample from a latent space to some distribution to increase the classification errors of the discriminator.

---

[1] In traditional machine learning scenarios, "unseen" data corresponds to data that is not used or seen during the training stage but rather the testing stage. The distribution of "unseen" data could be same as or different

Nevertheless, if a generator can learn to create unseen data, then a traditional GAN requires numerous training samples of unseen classes for training, leading to a contradiction with the definition of the unseen data. This fact motivates us to present the DSGAN, which can generate unseen data by adopting seen data as training samples (see Fig. 9, which illustrates the difference between GAN and the DSGAN, in Appendix A). The key concept is to consider the distribution of the unseen data as the difference between two distributions that are relatively easy to obtain. For example, the out-of-distribution examples in the MNIST dataset, from another perspective, are found to belong to the differences between the sets of examples in MNIST and the universal set. It should be noted that in traditional GAN, the target distribution is identical to the training data distribution; however, in the DSGAN these two distributions, are considered to be different.

This paper makes the following contributions:

(1) We propose the DSGAN to generate any unseen data only if the density of the target (unseen data) distribution is the difference between those of any two distributions, $p_{\bar{d}}$ and $p_d$.

(2) We show that the DSGAN possesses the flexibility to learn different target (unseen data) distributions in two key applications, semi-supervised learning and novelty detection. Specifically, for novelty detection, the DSGAN can produce boundary points around the seen data because this type of unseen data is easily misclassified. For semi-supervised learning, the unseen data are linear combinations of any labeled data and unlabeled data, excluding the labeled and unlabeled data themselves[2].

(3) The DSGAN yields results comparable to a semi-supervised learning but with a short training time and low memory consumption. In novelty detection, combining both the DSGAN and variational auto-encoder (VAE, Kingma & Welling (2014b)) methods achieve the state-of-the-art results.

## 2 PROPOSED METHOD-DSGAN

### 2.1 FORMULATION

We denote the generator distribution as $p_g$ and training data distribution as $p_d$, both in an $N$-dimensional space. Let $p_{\bar{d}}$ be the distribution decided by the user. For example, $p_{\bar{d}}$ can be the convolution of $p_d$ and normal distribution. Let $p_t$ be the target distribution that the user is interested in, and it can be expressed as

$$(1 - \alpha)p_t(x) + \alpha p_d(x) = p_{\bar{d}}(x), \tag{1}$$

where $\alpha \in [0, 1]$. Our method, the DSGAN, aims to learn $p_g$ such that $p_g = p_t$. Note that if the support set of $p_d$ belongs to that of $p_{\bar{d}}$, then there exists at least an $\alpha$ such that the equality in (1) holds. However, even if the equality does not hold, intuitively, the DSGAN attempts to learn $p_g$ such that $p_g(x) \sim \dfrac{p_{\bar{d}}(x) - \alpha p_d(x)}{1 - \alpha}$ with the constraint, $p_g(x) \geq 0$. Specifically, the generator will output samples located in the high-density areas of $p_{\bar{d}} - \alpha p_d$. Furthermore, we show that the DSGAN can learn $p_g$, whose support set is the difference between those of $p_{\bar{d}}$ and $p_d$ in Theorem 1.

First, we formulate the generator and discriminator in GANs. The inputs, $z$, of the generator are drawn from $p_z(z)$ in an $M$-dimensional space. The generator function, $G(z; \theta_g) : \mathbb{R}^M \to \mathbb{R}^N$, represents a mapping to the data space, where $G$ is a differentiable function with parameter $\theta_g$. The discriminator is defined as $D(x; \theta_d) : \mathbb{R}^N \to [0, 1]$, which outputs a single scalar. $D(x)$ can be considered as the probability that $x$ belongs to a class of the real data.

Similar to traditional GAN, we train $D$ to distinguish the real data from the fake data sampled from $G$. Concurrently, $G$ is trained to produce realistic data that can mislead $D$. However, in the DSGAN, the definitions of "real data" and "fake data" are different from those in traditional GAN. The samples from $p_{\bar{d}}$ are considered as real, but those from the mixture distribution between $p_d$ and $p_g$ are considered as fake. The objective function is defined as follows:

---

from the "seen" data, according to applications. In this paper, we focus on the scenario that the two distributions are different.

[2]The linear combination of any labeled data and unlabeled data probably belongs to the set of seen data (labeled data and unlabeled data), which contradicts the definition of unseen data. Thus, the samples generated by the DSGAN should not include the seen data themselves.

$$V(G, D) := \mathbb{E}_{x \sim p_{\bar{d}}(x)} [\log D(x)] + (1 - \alpha)\mathbb{E}_{z \sim p_z(z)} [\log (1 - D(G(z)))] + \alpha\mathbb{E}_{x \sim p_d(x)} [\log (1 - D(x))].$$
(2)

We optimize (2) by a min–max game between $G$ and $D$, i.e.,

$$\min_{G} \max_{D} V(G, D).$$

During the training procedure, an iterative approach, like traditional GAN, is to alternate between $k$ steps of training $D$ and one step of training $G$. In practice, minibatch stochastic gradient descent via backpropagation is used to update $\theta_d$ and $\theta_g$. Thus, for each $p_g$, $p_d$, and $p_{\bar{d}}$, $m$ samples are required for computing the gradients, where $m$ is the number of samples in a minibatch. The training procedure is illustrated in Algorithm 1 in Appendix A. The DSGAN suffers from the same drawbacks as traditional GAN, (*e.g.*, mode collapse, overfitting, and strong discriminator) so that the generator gradient vanishes. There are literature Salimans et al. (2016); Arjovsky & Bottou (2017); Miyato et al. (2018) focusing on dealing with the above problems, and such concepts can be readily combined with the DSGAN.

Li et al. (2017) and Reed et al. (2016) proposed an objective function similar to (2). Their goal was to learn the conditional distribution of training data. However, we aim to learn the target distribution, $p_t$, in Eq. (1), and not the training data distribution.

## 2.2 CASE STUDY ON VARIOUS UNSEEN DATA GENERATION

To achieve a more intuitive understanding about the DSGAN, we conduct several case studies on two-dimensional (2D) synthetic datasets and MNIST. In Eq. (1), $\alpha = 0.8$ is used.

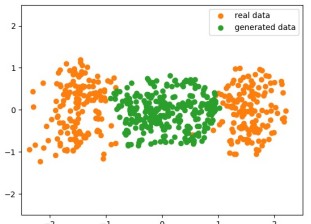

Figure 1: Complement points (in Green) between two circles (in Orange).

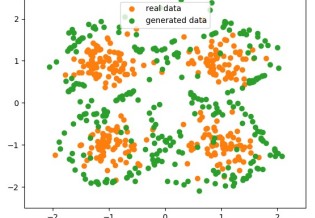

Figure 2: Boundary points (in Green) between four circles (in Orange).

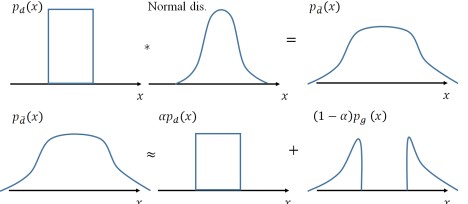

Figure 3: Illustration of the generation of the unseen data in the boundary around the training data. First, the convolution of $p_d$ and normal distribution ensure the density on the boundary is no longer zero. Second, we seek $p_g$ such that Eq. (1) holds, where the support set of $p_g$ is approximated by the difference of those between $p_{\bar{d}}$ and $p_d$.

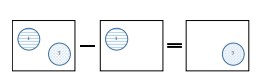

Figure 4: Illustration of the difference-set seeking in MNIST.

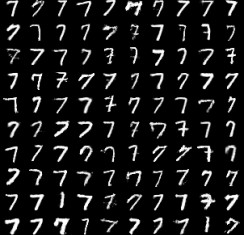

Figure 5: DSGAN learns the difference between two sets.

**Complement samples generation** Fig. 1 illustrates that the DSGAN can generate complement samples between 2 circles. Denoting the density function of the two circles as $p_d$, we assign the samples drawn from $p_{\bar{d}}$ as linear combinations of the two circles. Then, by applying the DSGAN, we achieve our goal of generating complement samples. In fact, this type of unseen data is used in semi-supervised learning.

**Boundary samples generation** Fig. 2 illustrates that the DSGAN generates boundary points between four circles. This type of unseen data is used in novelty detection. In this case, we assign $p_d$ and $p_{\bar{d}}$ as "the density function of four circles" and "the convolution of $p_d$ and normal distribution," respectively. The basis of our concept is also illustrated by a one-dimensional (1D) example in Fig. 3.

**Difference-set generation** We also validate the DSGAN on a high-dimensional dataset such as MNIST. In this example, we define $p_d$ as the distribution of digit "1" and $p_{\bar{d}}$ as the distribution containing two digits "1" and "7". Because the density, $p_d(x)$, is high when $x$ is digit "1," the generator is prone to output digit "7" with a high probability. More sample qualities of DSGAN on CelebA can be refer to Appendix G.

From the above results, we can observe two properties of the generator distribution, $p_g$: i) the higher the density of $p_d(x)$, the lower the density of $p_g(x)$; ii) $p_g$ prefers to output samples from the high-density areas of $p_{\bar{d}}(x) - \alpha p_d(x)$.

## 2.3   DESIGNING $p_{\bar{d}}$

Thus far, we have demonstrated how the DSGAN can produce various types of unseen data by choosing a specific $p_{\bar{d}}$. In this section, we introduce a standard procedure to design $p_{\bar{d}}$, and illustrate each step with pictures.

Step 1.   First, the training data, $p_d$, are collected (Fig. 6 (a)).

Step 2.   Second, based on the applications, the desired unseen data distribution is defined (*e.g.*, complement samples for semi-supervised learning) (Fig. 6 (b)).

Step 3.   Third, $p_{\bar{d}}$ is defined as a mixed distribution of $(1 - \alpha)p_g + (\alpha)p_d$ (Fig. 6 (c)).

Step 4.   Finally, a suitable mapping function that can transform $p_d$ to $p_{\bar{d}}$ is designed (*e.g*, linear combination of any two samples of $p_d$)

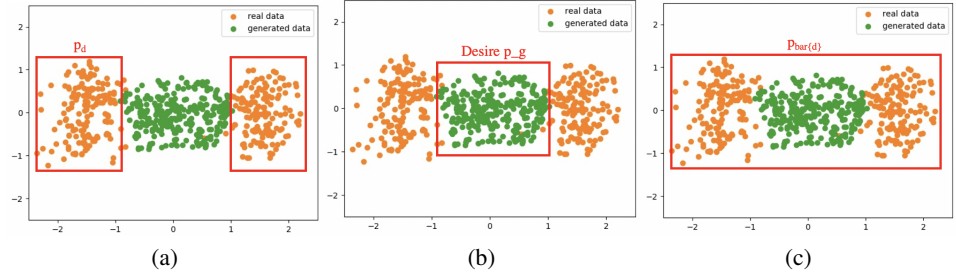

Figure 6: Illustration for designing $p_{\bar{d}}$.

In the above procedure, the most important step is to determine which types of unseen data are suitable for a specific problem (Step 2). In this paper, we show two types of unseen data, which are useful in semi-supervised learning and novelty detection. However, determining all types of unseen data for all applications is beyond the scope of this study, and we leave this for future work.

Furthermore, we provide a method (see Appendix B in supplementary materials) by reformulating the objective function (2), so that it is more stable to train the DSGAN.

## 3   THEORETICAL RESULTS

In this section, we show that by choosing an appropriate $\alpha$, the support set of $p_g$ belongs to the difference set between $p_{\bar{d}}$ and $p_d$, so that the samples from $p_g$ are unseen from the $p_d$ perspective.

We start our proofs from two assumptions. First, in a non-parametric setting, we assume that both the generator and discriminator have infinite capacities. Second, $p_g$ is defined as the distribution of the samples drawn from $G(z)$ under $z \sim p_z$.

In the following, we show that the support set of $p_g$ is contained within the differences in the support sets of $p_{\bar{d}}$ and $p_d$ while achieving the global minimum such that we can generate the desired $p_g$ by designing an appropriate $p_{\bar{d}}$.

**Theorem 1.** *Suppose $\alpha p_d(x) \geq p_{\bar{d}}(x)$ for all $x \in \text{Supp}(p_d)$ and all density functions $p_d(x)$, and $p_{\bar{d}}(x)$ and $p_g(x)$ are continuous. If the global minimum of $C(G)$ is achieved, then*

$$\text{Supp}(p_g) \subseteq \text{Supp}(p_{\bar{d}}) - \text{Supp}(p_d),$$

*where*

$$C(G) = \max_D V(G, D)$$

$$= \mathbb{E}_{x \sim p_{\bar{d}}(x)} \left[ \log \frac{p_{\bar{d}}(x)}{p_{\bar{d}}(x) + (1 - \alpha)p_g(x) + \alpha p_d(x)} \right] + \mathbb{E}_{x \sim p^*(x)} \left[ \log \frac{(1 - \alpha)p_g(x) + \alpha p_d(x)}{p_{\bar{d}}(x) + (1 - \alpha)p_g(x) + \alpha p_d(x)} \right].$$

*Proof.* See Appendix C for the details. $\square$

Summarizing, the generator is prone to output samples that are located in the high-density areas of $p_{\bar{d}} - \alpha p_d$.

## 4 APPLICATIONS

The DSGAN was applied to two problems: semi-supervised learning and novelty detection. In the semi-supervised learning, the DSGAN acts as a "bad generator," which creates complement samples (unseen data) in the feature space of the training data. For the novelty detection, the DSGAN generates the samples (unseen data) as boundary points around the training data.

### 4.1 SEMI-SUPERVISED LEARNING

Semi-supervised learning (SSL) is a type of learning model that uses a few labeled data and numerous unlabeled data. The existing SSL methods based on a generative model, (*e.g.*, VAE Kingma et al. (2014) and GAN Salimans et al. (2016)), yield good empirical results. Dai et al. (2017) theoretically showed that a good semi-supervised learning required a bad GAN with the following objective function:

$$\max_D \mathbb{E}_{x,y \sim \mathcal{L}} \log P_D(y \mid x, y \leq K) + \mathbb{E}_{x \sim p_d(x)} \log P_D(y \leq K \mid x) + \mathbb{E}_{x \sim p_g(x)} \log P_D(K + 1 \mid x), \quad (3)$$

where $(x, y)$ denotes a pair of data, and its corresponding label, $\{1, 2, \ldots, K\}$ denotes the label space for the classification, and $\mathcal{L} = \{(x, y)\}$ is the label dataset. Moreover, under the semi-supervised settings, $p_d$ in (3) is the distribution of the unlabeled data. Note that the discriminator, $D$, in GAN also plays the role of a classifier. If the generator distribution exactly matches the real data distribution (*i.e.*, $p_g = p_d$), then the classifier trained by the objective function (3) with the unlabeled data cannot have a better performance than that trained by the supervised learning with the objective function. Specifically,

$$\max_D \mathbb{E}_{x,y \sim \mathcal{L}} \log P_D(y \mid x, y \leq K). \quad (4)$$

Contrastingly, the generator is preferred to generate complement samples, which lie on the low-density area of $p_d$. Under some mild assumptions, these complement samples help $D$ to learn the correct decision boundaries in the low-density area because the probabilities of the true classes are forced to be low in the out-of-distribution areas.

The complement samples in Dai et al. (2017) are complex to produce. In Sec. 5.2, we will demonstrate that with the DSGAN, complement samples can be easily generated.

### 4.2 NOVELTY DETECTION

Novelty detection determines if a query example belongs to a seen class. If the samples of one seen class are considered as positive data, then this difficulty is the absence of negative data in the training phase, so that the supervised learning cannot function.

Recently, novelty detection has made significant progress with the advent of deep leaning. Pidhorskyi et al. (2018)Sakurada & Yairi (2014) focused on learning a representative latent space for a seen class. When testing, the query image was projected onto the learned latent space. Then, the difference between the query image and its inverse image (reconstruction) was measured. Thus, only an encoder was needed to be trained for the projection and a decoder for the reconstruction. Under the circumstance, an autoencoder (AE) is generally is adopted to learn both the encoder and decoder Pidhorskyi et al. (2018)Perera et al. (2019). Let $\text{Enc}(\cdot)$ be the encoder and $\text{Dec}(\cdot)$ be the decoder. The loss function of the AE is defined as

$$\min_{\text{Enc},\text{Dec}} \mathbb{E}_{x \sim p_{pos}(x)} \left[ \|x - \text{Dec}(\text{Enc}(x))\|_2^2 \right], \quad (5)$$

where $p_{pos}$ is the distribution of a seen class. After the training, a query example, $x_{test}$, is classified as the seen class if

$$\|x_{test} - \text{Dec}(\text{Enc}(x_{test}))\|_2^2 \leq \tau, \tag{6}$$

where $\tau \in \mathbb{R}^+$ plays the trade-off between the true positive rate and false positive rate. However, (6) is based on two assumptions: (1) the positive samples from one seen class should have a small reconstruction error; (2) the AE (or latent space) cannot well describe the negative examples from the unseen classes, leading to a relatively large reconstruction error. In general, the first assumption inherently holds when both the testing and training data originate from the same seen class. However, Pidhorskyi et al. (2018)Perera et al. (2019) observed that assumption (2) does not hold at all times because the loss function in (5) does not include a loss term to enforce the negative data to have a large reconstruction error.

For assumption (2) to hold, given positive data as the training inputs, we propose using the DSGAN to generate negative examples in the latent space, as discussed in Sec. 5.3. Then, the loss function of the AE is modified to enforce the negative data to have a large reconstruction error.

## 5 EXPERIMENTS

Our experiments are divided into three parts. The first one examines how the hyperparameter, $\alpha$, influences the learned generator distribution, $p_g$. In the second and third experiments, we obtain empirical results about semi-supervised learning and novelty detection, which are presented in Sec. 5.2 and Sec. 5.3, respectively. Note that the training procedure of the DSGAN can be improved by other extensions of GANs such as WGAN Arjovsky et al. (2017), WGAN-GP Gulrajani et al. (2017), EBGAN Zhao et al. (2017), and LSGAN Mao et al. (2017). In our method, the WGAN-GP was adopted for the stability of the DSGAN in training and reduction in the mode collapse.

### 5.1 DSGAN WITH DIFFERENT $\alpha$

The impacts of different $\alpha$ values on the DSGAN are illustrated in Fig. 7. In this example, the support of $p_d$ is the area bounded by a red dotted line, and the orange points are the samples from $p_d$. Concurrently, we shift $p_d$ to the right by 1 unit and create the distribution, $p_{\bar{d}}$, whose support is bounded by blue dotted lines. The overlapping area between $p_{\bar{d}}$ and $p_d$ is 0.5 unit (assuming the area of $p_d$ is 1 unit). Based on our theoretical results, $\alpha = 0.5$ is the smallest selected value allowing $p_g$ to be disjoint to $p_d$. Therefore, we can see that some generated samples, as presented in Fig. 7(a), still belong to the support set of $p_d$. Fig. 7(b) shows that there is a perfect agreement between our theoretical and experiment results with $\alpha = 0.5$. When $\alpha = 0.8$, there is a remarkable gap between the generated (green) points and yellow points, as shown in Fig. 7(c). In theory, the result obtained at $\alpha = 0.8$ should be the same as that obtained at $\alpha = 0.5$. This is because the discriminator should assign the entire area, which is the intersection of the complement of support set of $p_d$ and support set of $p_{\bar{d}}$, to the same score, under the assumption that the discriminator has an infinite capacity. However, in practice, the capacity of the discriminator is limited. Therefore, the score of the area near $p_d$ is lower than that far from it, when $\alpha$ is large. Therefore, $p_g$ tends to repel $p_d$ to achieve a high score (to deceive the discriminator).

### 5.2 DSGAN IN SEMI-SUPERVISED LEARNING

We first introduce how the DSGAN generates the complement samples in the feature space. Dai et al. (2017) proved that if the complement samples generated by $G$ could satisfy the following two assumptions in (7) and (8), i.e.,

$$\forall x \sim p_g(x), 0 > \max_{1 \leq i \leq K} w_i^T f(x) \text{ and } \forall x \sim p_d(x), 0 < \max_{1 \leq i \leq K} w_i^T f(x), \tag{7}$$

where $f$ is the feature extractor and $w_i$ is the linear classifier for the $i^{th}$ class, and

$$\begin{aligned} &\forall x_1 \sim \mathcal{L}, x_2 \sim p_d(x), \exists x_g \sim p_g(x) \text{ s.t.} \\ &f(x_g) = \beta f(x_1) + (1 - \beta) f(x_2) \text{ with } \beta \in [0, \ 1], \end{aligned} \tag{8}$$

then all the unlabeled data would be correctly classified by the objective function (3). Specifically, (7) ensures that the classifiers can discriminate the generated data from the unlabeled data, and (8) causes the decision boundary to be located in the low-density areas of $p_d$.

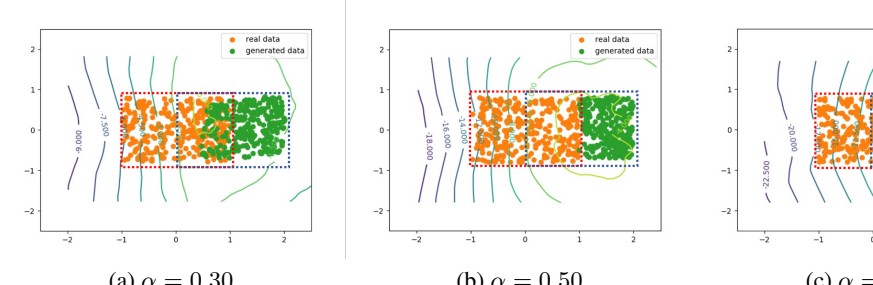

$$(a)\ \alpha = 0.30 \qquad\qquad (b)\ \alpha = 0.50 \qquad\qquad (c)\ \alpha = 0.80$$

Figure 7: Influence of $\alpha$ on the synthetic dataset. We observe that the samples of $p_g$ (green points) move farther away from $p_d$ as $\alpha$ increases; however, they are still bounded by the support of $p_{\bar{d}}$. When $\alpha$ is 0.5, the support set of $p_g$ is disjoint to that of $p_d$, satisying the theoretical results. When $\alpha$ is 0.8, $p_g$ generates the rightmost points of $p_{\bar{d}}$. The level curves from the discriminator show that the generator is more prone to producing samples in a region with higher score than in that with a lower score. Note that the outputs of the discriminator are not restricted in $[0, 1]$, because we use the formulation of the WGAN in this experiment.

The assumption in (8) implies that the complement samples must be in the space created by the linear combination of the labeled and unlabeled data. In addition, they cannot fall into the real data distribution, $p_d$, owing to the assumption (7). To allow the DSGAN to generate such samples, we let the samples of $p_{\bar{d}}$ be linear combinations of those from $\mathcal{L}$ and $p_d$. Since $p_g(x) \approx \dfrac{p_{\bar{d}}(x) - \alpha p_d(x)}{1 - \alpha}$, $p_g$ will tend to match $p_{\bar{d}}$, whereas the term, $-\alpha p_d$, ensures that the samples from $p_g$ do not belong to $p_d$. Thus, $p_g$ satisfies the assumption in (8). Moreover, (7) is also satisfied by training the classifier with (3) based on substituting the generator distribution in (3) into the learned $p_g$.

Following the previous works, we apply the proposed DSGAN to semi-supervised learning on three benchmark datasets: MNIST LeCun et al. (1998), SVHN Netzer et al. (2011), and CIFAR-10 Krizhevsky (2009). The details of the experiments can be found in Appendix D.

### 5.2.1 SIMULATION RESULTS

First, the selected hyperparameters are listed in Table 5 in Appendix D.1. Second, the results obtained from the DSGAN and state-of-the-art methods on the three benchmark datasets are summarized in Table 1. It can be observed that our method can compete with the state-of-the-art methods on the three datasets. Note that we report the results of badGAN not only from the original papers in the literature but also by reproducing them using the released codes of the authors. The reason of presenting both the results is that we cannot reproduce parts of the results. The experiments in Li et al. (2019) also showed a similar problem. In comparison with Dai et al. (2017), our methods do not need to rely on an additional density estimation network, PixelCNN++ Salimans et al. (2017). Although PixelCNN++ is one of the best density estimation networks, learning such a deep architecture requires large computation and high memory consumption. In Table 2, we list the training time and memory consumption for our method and badGAN. Compared to badGAN, our method consumes $15.8\%$ less training time and saves about 9000 MB during the training.

Moreover, it can also be observed from Table 1 that our results are comparable to the best record of badGAN and CAGAN. are better than those of other approaches on the MNIST and SVHN datasets. On CIFAR-10, our method is only inferior to the CT-GAN. However, this might not be a reasonable comparison because the CT-GAN uses extra techniques, including temporal ensembling and data augmentation, which the other methods do not use.

### 5.3 DSGAN IN NOVELTY DETECTION

In this section, we study how to use the DSGAN for assisting novelty detection. As mentioned in Sec. 4.2, we need to train the auto-encoder (AE) such that (i) the positive samples from one seen class have a small reconstruction error; (ii) negative samples from the unseen classes incur relatively higher reconstruction errors.

Table 1: Comparison of the semi-supervised learning in our DSGAN and the state-of-the-art methods. For a reasonable comparison, we only consider GAN-based methods. $*$ denotes the use of the same architecture of the classifier. $\dagger$ denotes a larger architecture of the classifier. $\ddagger$ denotes the use of data augmentation (in CIFAR-10). The results for MNIST are recorded as the number of errors, whereas for the others are as percentage of the error.

| Methods | MNIST | SVHN | CIFAR-10 |
|---------|-------|------|----------|
| FM$^*$ Salimans et al. (2016) | $93 \pm 6.5$ | $8.11 \pm 1.3$ | $18.63 \pm 1.32$ |
| TripleGAN$^\dagger$ Li et al. (2017) | $91 \pm 58$ | $5.77 \pm 0.17$ | $16.99 \pm 0.36$ |
| badGAN$^*$ Dai et al. (2017) | $79.5 \pm 9.8$ | $4.25 \pm 0.03$ | $14.41 \pm 0.30$ |
| CAGAN$^*$ Ni et al. (2018) | $81.9 \pm 4.5$ | $4.83 \pm 0.09$ | $12.61 \pm 0.12$ |
| CT-GAN$^\ddagger$ Wei et al. (2018) | $89 \pm 13$ | - | $9.98 \pm 0.21$ |
| badGAN-reproduce$^*$ | $86.2 \pm 13.2$ | $4.48 \pm 0.16$ | $16.25 \pm 0.33$ |
| Our method$^*$ | $82.7 \pm 4.6$ | $4.38 \pm 0.10$ | $14.52 \pm 0.14$ |

Table 2: Training times of our method and badGAN. We only report the training time on MNIST, on which the authors of badGAN applied PixelCNN++. The experiments run on a NVIDIA 1080 Ti.

| Methods | Training time | Memory Consumption |
|---------|---------------|--------------------|
| badGAN | 38 s / epoch | 9763 MB |
| Our method$^*$ | 32 s / epoch | 711 MB |

The fundamental concept is to use the DSGAN to generate negative samples, which originally do not exist under the scenario of novelty detection. Next, we add a new loss term to penalize the small reconstruction errors of the negative samples (see the third stage below). Three stages are required to train our model (AE):

1. The encoder, $\mathrm{Enc}(\cdot)$, and decoder, $\mathrm{Dec}(\cdot)$, are trained using the loss function (5).

2. Given $x \sim p_{pos}$, $\mathrm{Enc}(x)$ are collected as the samples drawn from $p_d$. $p_{\bar{d}}$ is the convolution of $p_d$ having a normal distribution with a zero mean and variance $\sigma$. Then, we train the DSGAN to generate negative samples, which are drawn from $p_{\bar{d}}(x) - p_d(x)$ and are the boundary points around the positive samples in the latent space. Note that there are some variations in the DSGAN: the input of the generator, $G$, is $\mathrm{Enc}(x)$, instead of a random vector $z$ in the latent space. We also add $\| \mathrm{Enc}(x) - G(\mathrm{Enc}(x)) \|_2^2$, which will be explained in the next step, to train the generator.

3. Fixing the encoder, we retrain the decoder by the modified loss function,

$$\min_{\mathrm{Dec}} \mathbb{E}_{x \sim p_{pos}(x)} \left[ \|x - \mathrm{Dec}(\mathrm{Enc}(x))\|_2^2 + w \cdot \max\left(0, m - \|x - \mathrm{Dec}(G(\mathrm{Enc}(x)))\|_2^2\right) \right],$$

where $w$ is the trade-off between the reconstruction errors of positive samples $\mathrm{Enc}(x)$ and negative samples $G(\mathrm{Enc}(x))$. Note that in the previous step, we add $\| \mathrm{Enc}(x) - G(\mathrm{Enc}(x)) \|_2^2$ to ensure that the outputs of the generator are around the input. Thus, the second term charges even though the negative samples are close to the corresponding positive sample, and they still exhibit a high reconstruction error, which is bounded by $m$ (Zhao et al. (2017)).

The above algorithm, called VAE+DSGAN, can be used to strengthen the existing AE-based methods by using them in the first stage. In the simulation, we used a variational autoencoder (VAE) Kingma & Welling (2014a) because it performs better than the AE in the novelty detection.

### 5.3.1 SIMULATION RESULTS

In this section, following Perera et al. (2019), the performance was evaluated using the area under the curve (AUC) of the receiver operating characteristics (ROC) curve. Given a dataset, a class was chosen as the seen class for training, and all the classes were used for testing. There exist several testing benchmarks for novelty detection, such as MNIST, COIL100 Nene et al. (1996) and CIFAR-10. The state-of-the-art method Perera et al. (2019) achieves high performance in AUC on MNIST and COIL100 (AUC is larger than 0.97). However, for CIFAR-10, Perera et al. (2019) only

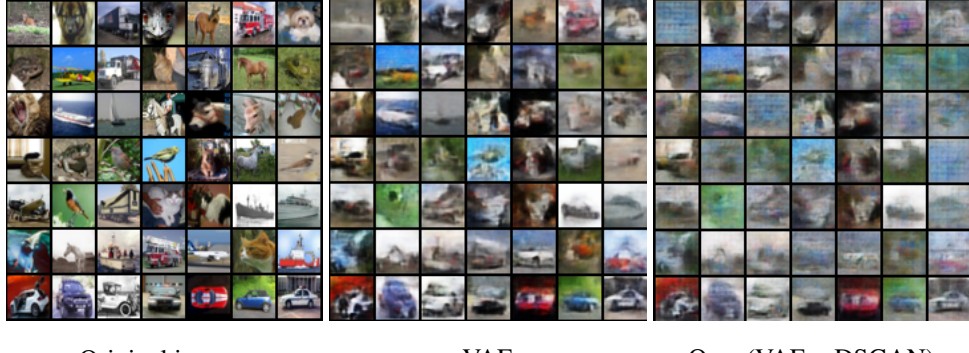

| Original images | VAE | Ours (VAE + DSGAN) |

Figure 8: Comparison of the reconstructed results of the VAE and our method. The seen class, which is at the bottom of the images, is a car. Other rows are images from the unseen classes. Our method exhibits a relatively larger gap, in terms of the reconstruction error between the seen data and unseen data, than the VAE.

Table 3: Comparison of our method (VAE+DSGAN) and the state-of-the-art methods: VAE Kingma & Welling (2014a), AND Abati et al. (2019), DSVDD Ruff et al. (2018), and OCGAN Perera et al. (2019). The results for CIFAR-10 are recorded in terms of the AUC value. The number in the top row denotes the seen class, where 0: Plain, 1: Car, 2: Bird, 3: Cat, 4: Deer, 5: Dog, 6: Frog,7: Horse, 8: Ship, 9: Truck.

|  | 0 | 1 | 2 | 3 | 4 | 5 | 6 | 7 | 8 | 9 | MEAN |
|---|---|---|---|---|---|---|---|---|---|---|---|
| VAE | .700 | .386 | .679 | .535 | .748 | .523 | .687 | .493 | .696 | .386 | .583 |
| AND | .735 | .580 | **.690** | .542 | **.761** | .546 | .751 | .535 | .717 | .548 | .641 |
| DSVDD | .617 | **.659** | .508 | .591 | .609 | **.657** | .677 | **.673** | .759 | **.731** | .648 |
| OCGAN | **..757** | .531 | .640 | .620 | .723 | .620 | **.723** | .575 | **.820** | .554 | .657 |
| Our method | .737 | .614 | .676 | **.644** | .759 | .562 | .660 | .646 | .769 | .633 | **.670** |

achieves 0.656. Thus, we chose the challenging dataset, CIFAR-10, as the benchmark to evaluate our method. The detailed network architecture can be found in Appendix E.

Because VAE+DSGAN can be considered as a fine tuning VAE Kingma & Welling (2014a), we first illustrate the key difference between the VAE and VAE+DSGAN, as shown in Fig. 8. The seen class, which is at the bottom of the images, is a car. Other rows are the images from the unseen classes. One can see that the reconstructed images are reasonably good even for the unseen class in the VAE. By contrast, our method enforces the reconstructed images of the unseen classes to be blurred while still preserving the reconstruction quality of the seen class. Thus, our method achieves a relatively larger gap, in terms of the reconstruction error between the seen data and unseen data, than the VAE.

In Table 3, we compare the proposed method with several methods, including the VAE Kingma & Welling (2014a), AND Abati et al. (2019), DSVDD Ruff et al. (2018), and OCGAN Perera et al. (2019), in terms of the AUC value. One can see that in most cases, our method almost outperforms the VAE. Furthermore, the mean of the AUC values of our method also is larger than those of the state-of-the-art methods. It is worth mentioning that in addition to the VAE, the DSGAN has potential of being combined with other AE-based methods.

## 6 RELATED WORKS ABOUT UNSEEN DATA GENERATION

Yu et al. (2017) proposed a method to generate samples of unseen classes in a unsupervised manner via an adversarial learning strategy. However, it requires solving an optimization problem for each sample, which certainly leads to a high computation cost. By contrast, the DSGAN has the capability to create infinite diverse unseen samples. Hou et al. (2018) presented a new GAN architecture that could learn two distributions of unseen data from a part of seen data and the unlabeled data. However, the unlabeled data must be a mixture of seen and unseen samples; the DSGAN does not require any unseen data. Kliger & Fleishman (2018) also applied GAN in novelty detection. Their objective

was to learn a generator whose distribution is a mixture of novelty data distribution and training data distribution. To this end, they used feature matching (FM) to train the generator and expected $p_g$ to learn the mixture of distributions. However, the ultimate goal of FM is still to learn $p_g = p_d$; therefore, their method might fail when GAN learns well.

Dai et al. (2017) aimed to generate complementary samples (or out-of-distribution samples), but assumed that the in-distribution could be estimated by a pre-trained model, such as PixelCNN++, which might be difficult and expensive to train. Lee et al. (2018) used a simple classifier to replace the role of PixelCNN++ in Dai et al. (2017) so that the training was comparatively much easier and more suitable. Nevertheless, their method only focused on generating unseen data surrounding the low-density area of seen data. In comparison, the DSGAN has more flexibility to generate different types of unseen data (*e.g.,* a linear combination of seen data, as described in Sec. 5.2). In addition, their method needs the label information of the data, whereas our method is fully unsupervised.

## 7 CONCLUSIONS

We propose the DSGAN, which can produce any unseen data based on the assumption that the density of the unseen data distribution is the difference between the densities of any two distributions. The DSGAN is useful in an environment when the samples from the unseen data distribution are more difficult to collect than those from the two known distributions. Empirical and theoretical results are provided to validate the effectiveness of the DSGAN. Finally, because the DSGAN is developed based on GAN, it is easy to apply any improved versions of GAN to the DSGAN.

## 8 ACKNOWLEDGEMENT

This work was partially supported by grants MOST 107-2221-E-001-015-MY2 and MOST 108-2634-F-007-010 from Ministry of Science and Technology, Taiwan, ROC.

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

APPENDIX

## A   FLOWCHART AND ALGORITHM OF DSGAN

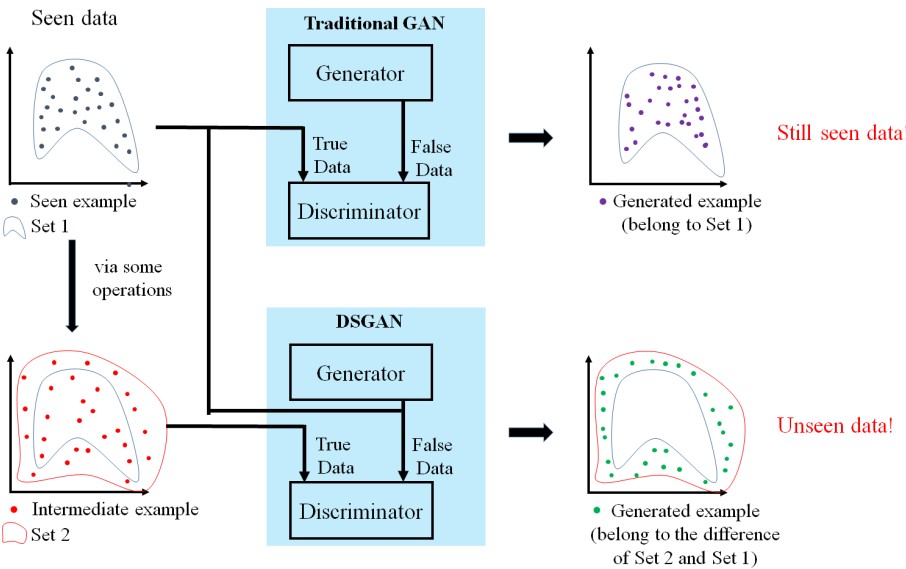

Figure 9: Illustration of the differences between traditional GAN and DSGAN.

**Algorithm 1** The training procedure of DSGAN using minibatch stochastic gradient descent. $k$ is the number of steps applied to discriminator. $\alpha$ is the ratio between $p_g$ and $p_d$ in the mixture distribution. We used $k = 1$ and $\alpha = 0.8$ in experiments.

01.   **for** number of training iterations **do**
02.      **for** $k$ steps **do**
03.         Sample minibatch of $m$ noise samples $z^{(1)}, ..., z^{(m)}$ from $p_g(z)$.
04.         Sample minibatch of $m$ samples $x_d^{(1)}, ..., x_d^{(m)}$ from $p_d(x)$.
05.         Sample minibatch of $m$ samples $x_{\bar{d}}^{(1)}, ..., x_{\bar{d}}^{(m)}$ from $p_{\bar{d}}(x)$.
06.         Update the discriminator by ascending its stochastic gradient:

$$\nabla_{\theta_d} \left[ \frac{1}{m} \sum_{i=1}^{m} \log D\left(x_d^{(i)}\right) + \log\left(1 - D\left(G\left(z^{(i)}\right)\right)\right) + \log\left(1 - D\left(x_{\bar{d}}^{(i)}\right)\right) \right]$$

07.      **end for**
08.      Sample minibatch of $m$ noise samples $z^{(1)}, ..., z^{(m)}$ from $p_g(z)$.
09.      Update the generator by descending its stochastic gradient:

$$\nabla_{\theta_g} \frac{1}{m} \sum_{i=1}^{m} \left[ \log\left(1 - D\left(G\left(z^{(i)}\right)\right)\right) \right]$$

10.   **end for**

## B    TRICKS FOR STABLE TRAINING

We provide a trick to stabilize the training procedure by reformulating the objective function. Specifically, $V(G, D)$ in (2) is reformulated as:

$$
\begin{aligned}
V(G, D) &= \int_x p_{\bar{d}}(x) \log(D(x)) \\
&\quad + ((1 - \alpha)p_g(x) + \alpha p_d(x)) \log(1 - D(x)) \, dx \\
&= \mathbb{E}_{x \sim p_{\bar{d}}(x)} [\log D(x)] \\
&\quad + \mathbb{E}_{x \sim (1-\alpha)p_g(x) + \alpha \sim p_d(x)} [\log(1 - D(x))].
\end{aligned}
\tag{9}
$$

Instead of sampling a mini-batch of $m$ samples from $p_z$ and $p_d$ in Algorithm 1, $(1 - \alpha)m$ and $\alpha m$ samples from both distributions are required, respectively. The computation cost in training can be reduced due to fewer samples. Furthermore, although (9) is equivalent to (2) in theory, we find that the training using (9) achieves better performance than using (2) via empirical validation in Table 4. We conjecture that the equivalence between (9) and (2) is based on the linearity of expectation, but mini-batch stochastic gradient descent in practical training may lead to the different outcomes.

Table 4: Semi-supervised learning results on MNIST with and without the use of sampling tricks.

| Methods | MNIST (# errors) |
|---|---|
| Our method w/o tricks | $91.0 \pm 7.0$ |
| Our method w/ tricks | $82.7 \pm 4.6$ |

## C    PROOF OF THEOREM 1

In this section, we show Theorem 1.

This proof includes two parts: the first part shows that the objective function is equivalent to minimizing the Jensen–Shannon divergence in the mixture distribution ($p_d$ and $p_g$) and $p_{\bar{d}}$ if $G$ and $D$ are assigned sufficient capacity; the second part shows that by choosing an appropriate $\alpha$, the support set of $p_g$ belongs to the difference set between $p_{\bar{d}}$ and $p_d$, so that the samples from $p_g$ are unseen from the $p_d$ perspective.

For the first part, we show the optimal discriminator given $G$, and then show that minimizing $V(G, D)$ via $G$, given the optimal discriminator, is equivalent to minimizing the Jensen–Shannon divergence between $(1 - \alpha)p_g + \alpha p_d$ and $p_{\bar{d}}$.

**Proposition 1.** *If $G$ is fixed, the optimal discriminator, $D$, is*

$$
D_G^*(x) = \frac{p_d(x)}{p_d(x) + (1 - \alpha)p_g(x) + \alpha p_d(x)}.
$$

*Proof.* Given any generator $G$, the training criterion for the discriminator $D$ is to maximize the quantity $V(G, D)$:

$$V(G, D) = \int_x p_{\bar{d}}(x) \log(D(x)) \, dx$$

$$+ (1 - \alpha) \int_z p_z(z) \log(1 - D(G(z))) \, dz$$

$$+ \alpha \int_x p_d(x) \log(1 - D(x)) \, dx$$

$$= \int_x p_{\bar{d}}(x) \log(D(x)) \, dx$$

$$+ (1 - \alpha) \int_x p_g(x) \log(1 - D(x)) \, dz$$

$$+ \alpha \int_x p_d(x) \log(1 - D(x)) \, dx$$

$$= \int_x p_{\bar{d}}(x) \log(D(x))$$

$$+ ((1 - \alpha)p_g(x) + \alpha p_d(x)) \log(1 - D(x)) \, dx.$$

For any $(a, b) \in \mathbb{R}^2 \backslash \{0, 0\}$, the function $a \log(y) + b \log(1 - y)$ achieves its maximum in $[0, 1]$ at $y = \frac{a}{a+b}$. The discriminator only needs to be defined within $\text{Supp}(p_{\bar{d}}) \bigcup \text{Supp}(p_d) \bigcup \text{Supp}(p_g)$. We complete this proof. $\qquad \square$

Moreover, $D$ can be considered to discriminate between samples from $p_{\bar{d}}$ and $((1 - \alpha)p_g(x) + \alpha p_d(x))$. By replacing the optimal discriminator in $V(G, D)$, we trivially obtain

$$C(G) = \max_D V(G, D)$$

$$= \mathbb{E}_{x \sim p_{\bar{d}}(x)} \left[ \log \frac{p_{\bar{d}}(x)}{p_{\bar{d}}(x) + (1 - \alpha)p_g(x) + \alpha p_d(x)} \right] + \mathbb{E}_{x \sim p^*(x)} \left[ \log \frac{(1 - \alpha)p_g(x) + \alpha p_d(x)}{p_{\bar{d}}(x) + (1 - \alpha)p_g(x) + \alpha p_d(x)} \right]. \tag{10}$$

Actually, the results thus far yield the optimal solution of $D$ given $G$ is fixed in (1). Now, the next step is to determine the optimal $G$ with $D_G^*$ as fixed.

**Theorem 2.** *The global minimum of $C(G)$ is achieved if and only if $(1 - \alpha)p_g(x) + \alpha p_d(x) = p_{\bar{d}(x)}$ for all $x$. Then, $C(G)$ achieves the value, $- \log 4$.*

*Proof.* We start from

$$(1) = - \log(4)$$

$$+ \mathbb{E}_{x \sim p_{\bar{d}}(x)} \left[ \log \frac{2p_{\bar{d}}(x)}{p_{\bar{d}}(x) + (1 - \alpha)p_g(x) + \alpha p_d(x)} \right]$$

$$+ \mathbb{E}_{x \sim p^*(x)} \left[ \log \frac{2((1 - \alpha)p_g(x) + \alpha p_d(x))}{p_{\bar{d}}(x) + (1 - \alpha)p_g(x) + \alpha p_d(x)} \right]$$

$$= - \log(4) + \text{KL} \left( p_{\bar{d}} \, \middle\| \, \frac{p_{\bar{d}} + (1 - \alpha)p_g + \alpha p_d}{2} \right)$$

$$+ \text{KL} \left( (1 - \alpha)p_g(x) + \alpha p_d \, \middle\| \, \frac{p_{\bar{d}} + (1 - \alpha)p_g + \alpha p_d}{2} \right)$$

$$= - \log(4) + 2 \, \text{JSD} \left( p_{\bar{d}} \, \| \, (1 - \alpha)p_g + \alpha p_d \right),$$

where $p^*(x) = (1 - \alpha)p_g(x) + \alpha p_d(x)$, KL is the Kullback-Leibler divergence and JSD is the Jensen-Shannon divergence. The JSD returns the minimal value, which is 0, iff both distributions are the same, namely $p_{\bar{d}} = (1 - \alpha)p_g + \alpha p_d$. Because $p_g(x)$'s are always non-negative, it should be noted both distributions are the same only if $\alpha p_d(x) \leq p_{\bar{d}}(x)$ for all $x$'s. We complete this proof. $\quad \square$

Note that $(1-\alpha)p_g(x) + \alpha p_d(x) = p_{\bar{d}(x)}$ may not hold if $\alpha p_d(x) > p_{\bar{d}(x)}$. However, the DSGAN still works based on two facts: i) given $D$, $V(G, D)$ is a convex function in $p_g$ and ii) because $\int_x p_g(x)dx = 1$, the set collecting all the feasible solutions of $p_g$ is convex. Thus, there always exists a global minimum of $V(G, D)$ given $D$, but it may not be $-\log(4)$.

Now, we go back to prove Theorem 1. We show that the support set of $p_g$ is contained within the differences in the support sets of $p_{\bar{d}}$ and $p_d$ while achieving the global minimum such that we can generate the desired $p_g$ by designing an appropriate $p_{\bar{d}}$.

*Proof.* Recall that

$$
\begin{aligned}
C(G) &= \int_x p_{\bar{d}}(x) \log \left( \frac{p_{\bar{d}}(x)}{p_{\bar{d}}(x) + (1-\alpha)p_g(x) + \alpha p_d(x)} \right) \\
&\quad + p^*(x) \log \left( \frac{(1-\alpha)p_g(x) + \alpha p_d(x)}{p_{\bar{d}}(x) + (1-\alpha)p_g(x) + \alpha p_d(x)} \right) dx \\
&= \int_x S(p_g; x)dx \\
&= \int_{x \in \mathrm{Supp}(p_{\bar{d}}) - \mathrm{Supp}(p_d)} S(p_g; x)dx \\
&\quad + \int_{x \in \mathrm{Supp}(p_d)} S(p_g; x)dx.
\end{aligned}
$$

$S(p_g; x)$ is used to simplify the notations inside the integral. For any $x$, $S(p_g; x)$ in $p_g(x)$ is non-increasing and $S(p_g; x) \leq 0$ always holds. Specifically, $S(p_g; x)$ is decreasing along the increase of $p_g(x)$ if $p_{\bar{d}}(x) > 0$; $S(p_g; x)$ attains the maximum value, zero, for any $p_g(x)$ if $p_{\bar{d}}(x) = 0$. Since DSGAN aims to minimize $C(G)$ with the constraint $\int_x p_g(d)dx = 1$, the solution attaining the global minima must satisfy $p_g(x) = 0$ if $p_{\bar{d}}(x) = 0$; otherwise, there exists another solution with smaller value of $C(G)$. Thus, $\mathrm{Supp}(p_g) \subseteq \mathrm{Supp}(p_{\bar{d}})$.

Furthermore, $T(p_g; x) = \frac{\partial S(p_g; x)}{\partial p_g(x)} = \log \left( \frac{(1-\alpha)p_g(x) + \alpha p_d(x)}{p_{\bar{d}}(x) + (1-\alpha)p_g(x) + \alpha p_d(x)} \right)$, which is expected to be as small as possible to minimize $C(G)$, is increasing on $p_g(x)$ and converges to 0. Then, we show that $T(p_g; x)$ for $x \in \mathrm{Supp}(p_{\bar{d}}) \bigcap \mathrm{Supp}(p_d)$ is always larger than that for $x \in \mathrm{Supp}(p_{\bar{d}}) - \mathrm{Supp}(p_d)$ for all $p_g$. Specifically,

1. When $x \in \mathrm{Supp}(p_{\bar{d}}) \bigcap \mathrm{Supp}(p_d)$, $T(p_g; x) \geq \log \frac{1}{2}$ always holds due to the assumption of $\alpha p_d(x) \geq p_{\bar{d}}(x)$.

2. When $x \in \mathrm{Supp}(p_{\bar{d}}) - \mathrm{Supp}(p_d)$, $T(p_g; x) < \log \frac{1}{2}$ for all $p_g(x)$'s satisfying $(1-\alpha)p_g(x) \leq p_{\bar{d}(x)}$.

Thus, the minimizer prefers $p_g(x) > 0$ for $x \in \mathrm{Supp}(p_{\bar{d}}) - \mathrm{Supp}(p_d)$ and $(1-\alpha)p_g(x) \leq p_{\bar{d}(x)}$. We check whether there exists a solution $p_g$ such that $(1-\alpha)p_g(x) \leq p_{\bar{d}(x)}$ and $\int_{x \in \mathrm{Supp}(p_{\bar{d}}) - \mathrm{Supp}(p_d)} p_g(d)dx = 1$, implying $p_g(x) = 0$ for $x \in \mathrm{Supp}(p_{\bar{d}}) \bigcap \mathrm{Supp}(p_d)$. Based

on the following expression,

$$\int_{x\in\text{Supp}(p_{\bar{d}})-\text{Supp}(p_d)} p_{\bar{d}}(x)dx \;+\; \int_{x\in\text{Supp}(p_d)} p_{\bar{d}}(x)dx = 1$$

$$\Rightarrow \int_{x\in\text{Supp}(p_{\bar{d}})-\text{Supp}(p_d)} p_{\bar{d}}(x)dx$$

$$\geq\; 1 - \int_{x\in\text{Supp}(p_d)} \alpha p_d(x)dx$$

$$\Rightarrow \int_{x\in\text{Supp}(p_{\bar{d}})-\text{Supp}(p_d)} p_{\bar{d}}(x)dx \;\geq\; 1-\alpha$$

$$\Rightarrow \int_{x\in\text{Supp}(p_{\bar{d}})-\text{Supp}(p_d)} p_{\bar{d}}(x)dx$$

$$\geq\; \int_{x\in\text{Supp}(p_{\bar{d}})-\text{Supp}(p_d)} (1-\alpha)p_g(x)dx,$$

the last inequality implies that there must exist a feasible solution. We complete this proof. $\qquad\square$

Another concern is the convergence of Algorithm 1.

**Proposition 2.** *The discriminator reaches its optimal value given $G$ in Algorithm 1, and $p_g$ is updated by minimizing*

$$\mathbb{E}_{x\sim p_{\bar{d}}(x)}\left[\log D_G^*(x)\right] + \mathbb{E}_{x\sim p^*(x)}\left[\log\left(1-D_G^*(x)\right)\right].$$

*If $G$ and $D$ have sufficient capacities, then $p_g$ converges to $\underset{p_g}{\arg\min}\,\text{JSD}\left(p_{\bar{d}}\,\|\,(1-\alpha)p_g + \alpha p_d\right)$.*

*Proof.* Consider $V(G,D) = U(p_g, D)$ as a function of $p_g$. By the proof idea of Theorem 2 in Goodfellow et al. (2014), if $f(x) = \sup_{\alpha\in\mathcal{A}} f_\alpha(x)$ and $f_\alpha(x)$ is convex in $x$ for every $\alpha$, then $\partial f_\beta(x) \in \partial f$ if $\beta = \arg\sup_{\alpha\in\mathcal{A}} f_\alpha(x)$. In other words, if $\sup_D V(G,D)$ is convex in $p_g$, the subderivatives of $\sup_D V(G,D)$ includes the derivative of the function at the point, where the maximum is attained, implying the convergence with sufficiently small updates of $p_g$. We complete this proof. $\qquad\square$

## D   EXPERIMENTAL DETAILS FOR SEMI-SUPERVISED LEARNING

### D.0.1   DATASETS: MNIST, SVHN, AND CIFAR-10

For evaluating the semi-supervised learning task, we used 60000/ 73257/ 50000 samples and 10000/ 26032/ 10000 samples from the MNIST/ SVHN/ CIFAR-10 datasets for the training and testing, respectively. Under the semi-supervised setting, we randomly chose 100/ 1000/ 4000 samples from the training samples, which are the MNIST/ SVHN/ CIFAR-10 labeled datasets, and the amounts of the labeled data for all the classes are equal. Furthermore, our criterion to determine the hyperparameters is introduced in Appendix D.1, and the network architectures are described in Appendix D.2. We performed testing with 10/ 5/ 5 runs on MNIST/ SVHN/ CIFAR-10 based on the selected hyperparameters, and randomly selected the labeled dataset. The results were recorded as the mean and standard deviation of the number of errors from each run.

### D.1   HYPERPARAMETERS

The hyperparameters were chosen to make our generated samples consistent with the assumptions in (7) and (8). However, in practice, if we make all the samples produced by the generator following the assumption in (8), then the generated distribution is not close to the true distribution, even a large margin between them exists, which is not what we desire. So, in our experiments, we make a concession that the percentage of generated samples, which accords with the assumption, is around 90%. To meet this objective, we tune the hyperparameters. Table 5 shows our setting of hyperparameters, where $\beta$ is defined in (8).

Table 5: Hyperparameters in semi-supervised learning.

| Hyperparameters | MNIST | SVHN | CIFAR-10 |
|:---:|:---:|:---:|:---:|
| $\alpha$ | 0.8 | 0.8 | 0.5 |
| $\beta$ | 0.3 | 0.1 | 0.1 |

### D.2 ARCHITECTURE

In order to fairly compare with other methods, our generators and classifiers for MNIST, SVHN, and CIFAR-10 are same as in Salimans et al. (2016) and Dai et al. (2017). However, different from previous works that have only a generator and a discriminator, we design an additional discriminator in the feature space, and its architecture is similar across all datasets with only the difference in the input dimensions. Following Dai et al. (2017), we also define the feature space as the input space of the output layer of discriminators.

Compared to SVHN and CIFAR-10, MNIST is a simple dataset as it is only composed of fully connected layers. Batch normalization (BN) or weight normalization (WN) is used in every layer to stable training. Moreover, Gaussian noise is added before each layer in the classifier, as proposed in Rasmus et al. (2015). We find that the added Gaussian noise exhibits a positive effect for semi-supervised learning. The architecture is shown in Table 6.

Table 7 and Table 8 are models for SVHN and CIFAR-10, respectively, and these models are almost the same except for some implicit differences, *e.g.*, the number of convolutional filters and types of dropout. In these tables, given a dropping rate, "Dropout" denotes a normal dropout in that the elements of input tensor are randomly set to zero while Dropout2d is a dropout only applied on the channels to randomly zero all the elements.

Table 6: Network architectures for semi-supervised learning on MNIST. (GN: Gaussian noise)

| Generator $G$ | Discriminator $D$ | Classifier $C$ |
|:---|:---|:---|
| Input: $\boldsymbol{z} \in \mathbb{R}^{100}$ from unif(0, 1) | Input: 250 dimension feature | Input: $28 \times 28$ gray image |
| $100 \times 500$ FC layer with BN
Softplus
$500 \times 500$ FC layer with BN
Softplus
$500 \times 784$ FC layer with WN
Sigmoid | $250 \times 400$ FC layer
ReLU
$400 \times 200$ FC layer
ReLU
$200 \times 100$ FC layer
ReLU
$100 \times 1$ FC layer | GN, std = 0.3
$784 \times 1000$ FC layer with WN ,ReLU
GN, std = 0.5
$1000 \times 500$ FC layer with WN, ReLU
GN, std = 0.5
$500 \times 250$ FC layer with WN, ReLU
GN, std = 0.5
$250 \times 250$ FC layer with WN, ReLU
GN, std = 0.5
$250 \times 250$ FC layer with WN, ReLU

$250 \times 10$ FC layer with WN |

Furthermore, the training procedure alternates between $k$ steps of optimizing $D$ and one step of optimizing $G$. We find that $k$ in Algorithm 1 is a key role in the problem of mode collapse for different applications. For semi-supervised learning, we set $k = 1$ for all datasets.

## E EXPERIMENTAL DETAILS FOR NOVELTY DETECTION

The architecture of GAN and VAE are depicted in Table 9 and 10, respectively.

In the experiment, we first trained the VAE for 500 epochs and then we trained DSGAN for 500 epochs with $m = 1.5$ and $w = 0.5$. Third, we fixed the encoder and tuned the decoder with both positive and negative samples (generated by DSGAN) for 600 epochs.

Table 7: The architectures of generator and discriminator for semi-supervised learning on SVHN and CIFAR-10. $N$ was set to 128 and 192 for SVHN and CIFAR-10, respectively.

| Generator $G$ | Discriminator $D$ |
|---|---|
| Input: $z \in \mathbb{R}^{100}$ from unif(0, 1) | Input: $N$ dimension feature |
| $100 \times 8192$ FC layer with BN, ReLU
Reshape to $4 \times 4 \times 512$
$5 \times 5$ conv. transpose 256 stride = 2 with BN, ReLU
$5 \times 5$ conv. transpose 128 stride = 2 with BN, ReLU
$5 \times 5$ conv. transpose 3 stride = 2 with WN, Tanh | $N \times 400$ FC layer, ReLU
$400 \times 200$ FC layer, ReLU
$200 \times 100$ FC layer, ReLU
$100 \times 1$ FC layer |

Table 8: The architecture of classifiers for semi-supervised learning on SVHN and CIFAR-10. (GN: Gaussian noise; lReLU(leak rate): LeakyReLU(leak rate))

| Classifier $C$ for SVHN | Classifier $C$ for CIFAR-10 |
|---|---|
| Input: $32 \times 32$ RGB image | Input: $32 \times 32$ RGB image |
| GN, std = 0.05
Dropout2d, dropping rate = 0.15
$3 \times 3$ conv. 64 stride = 1 with WN, lReLU(0.2)
$3 \times 3$ conv. 64 stride = 1 with WN, lReLU(0.2)
$3 \times 3$ conv. 64 stride = 2 with WN, lReLU(0.2)
Dropout2d, dropping rate = 0.5
$3 \times 3$ conv. 128 stride = 1 with WN, lReLU(0.2)
$3 \times 3$ conv. 128 stride = 1 with WN, lReLU(0.2)
$3 \times 3$ conv. 128 stride = 2 with WN, lReLU(0.2)
Dropout2d, dropping rate = 0.5
$3 \times 3$ conv. 128 stride = 1 with WN, lReLU(0.2)
$1 \times 1$ conv. 128 stride = 1 with WN, lReLU(0.2)
$1 \times 1$ conv. 128 stride = 1 with WN, lReLU(0.2)
Global average Pooling

$128 \times 10$ FC layer with WN | GN, std = 0.05
Dropout2d, dropping rate = 0.2
$3 \times 3$ conv. 96 stride = 1 with WN, lReLU(0.2)
$3 \times 3$ conv. 96 stride = 1 with WN, lReLU(0.2)
$3 \times 3$ conv. 96 stride = 2 with WN, lReLU(0.2)
Dropout, dropping rate = 0.5
$3 \times 3$ conv. 192 stride = 1 with WN, lReLU(0.2)
$3 \times 3$ conv. 192 stride = 1 with WN, lReLU(0.2)
$3 \times 3$ conv. 192 stride = 2 with WN, lReLU(0.2)
Dropout, dropping rate = 0.5
$3 \times 3$ conv. 192 stride = 1 with WN, lReLU(0.2)
$1 \times 1$ conv. 192 stride = 1 with WN, lReLU(0.2)
$1 \times 1$ conv. 192 stride = 1 with WN, lReLU(0.2)
Global average Pooling

$192 \times 10$ FC layer with WN |

Table 9: The architectures of generator and discriminator in DSGAN for novelty detection.

| Generator $G$ | Discriminator $D$ |
|---|---|
| Input: 128 dimension feature | Input: 128 dimension feature |
| $128 \times 1024$ FC layer with BN, ReLU
$1024 \times 512$ FC layer with BN, ReLU
$512 \times 256$ FC layer with BN, ReLU
$256 \times 128$ FC layer | $128 \times 400$ FC layer, ReLU
$400 \times 200$ FC layer, ReLU
$200 \times 100$ FC layer, ReLU
$100 \times 1$ FC layer |

# F ABLATION STUDY ON DIFFERENT $\alpha$ VALUES FOR SEMI-SUPERVISED LEARNING

Fig. 7 shows how different $\alpha$ values influence DSGAN. The optimal $\alpha$ for DSGAN to generate "unseen" data depends on $p_{\bar{d}}$ and $p_d$. According to Fig. 7, we can figure out that DSGAN is prone to generating unseen data under a larger $\alpha$. Recall that Theorem 1 illustrates $\alpha$ should be expected to be as large as possible if both network $G$ and $D$ have infinite capacity. Though the networks never have the infinite capacity in real applications, a general rule is to pick a large $\alpha$ and force the complement data to be far from $p_d$, which is similar to the results in Sec. 5.1.

Table 10: The architectures of VAE for novelty detection.

| Encoder | Decoder |
|---|---|
| $5 \times 5$ conv. 32 stride = 2, with BN, lReLU(0.2) | $5 \times 5$ conv. transpose 128 stride = 2 with BN, lReLU(0.2) |
| $5 \times 5$ conv. 64 stride = 2, with BN, lReLU(0.2) | $5 \times 5$ conv. transpose 64 stride = 2 with BN, lReLU(0.2) |
| $5 \times 5$ conv. 128 stride = 2, with BN, lReLU(0.2) | $5 \times 5$ conv. transpose 32 stride = 2 with BN, lReLU(0.2) |
| (For mean) | $5 \times 5$ conv. transpose 3 stride = 2, Tanh |
| $4 \times 4$ conv. 128 stride = 1 | |
| (For std) | |
| $4 \times 4$ conv. 128 stride = 1 | |

Here, we conduct the experiments on different $\alpha$ under semi-supervised learning settings. From Sec. 4.1 and 5.2, badGAN already shows that, if the desired unseen data can be generated, then the classifier will put the correct decision boundary in the low-density area.

In Table 11, we demonstrate the classification results on $\alpha = 0.5$ and $\alpha = 0.8$, respectively. We can observe that the results generated at $\alpha = 0.8$ is better than those generated at $\alpha = 0.5$, meeting the above discussion. From our empirical observations, DSGAN is prone to generating unseen data at $\alpha = 0.8$, leading a better classifier.

Table 11: Ablation study of different $\alpha$ values for DSGAN in semi-supervised learning, where the result for MNIST is represented in terms of number of errors and the percentage of errors was used for other datasets.

| Methods | MNIST | SVHN | CIFAR-10 |
|---|---|---|---|
| DSGAN ($\alpha = 0.5$) | $91.5 \pm 5.6$ | $4.59 \pm 0.15$ | $14.52 \pm 0.14$ |
| DSGAN ($\alpha = 0.8$) | $82.7 \pm 4.6$ | $4.38 \pm 0.10$ | $14.47 \pm 0.15$ |

## G   SAMPLE QUALITY OF DSGAN ON CELEBA

We show one more experiment on CelebA (Liu et al. (2015)) to demonstrate DSGAN can work well even for complicated images. In this experiment, we generate the color images of size $64 \times 64$. Similar to our 1/7 experiments on the MNIST dataset, we let $p_{\bar{d}}$ be the distribution of face images with glasses and without glasses and let $p_d$ be the distribution of images without glasses. We validate DSGAN with $\alpha = 0.5$ and $\alpha = 0.8$, respectively. For $\alpha = 0.5$, we sample 10000 images with glasses and 10000 images without glasses from CelebA. When $\alpha$ is 0.8, we sample 40000 instead of 10000 images without glasses.

We also train GAN to verify the generated image quality of DSGAN. For fair comparison, GAN is trained under two kinds of settings. The first one is that GAN is only trained with the images with glasses. Second, it is pretrained with all images, and is finetuned with the images with glasses, namely transferring GANs in Wang et al. (2018). It should be noted that transferring GAN uses the same amount of training data as DSGAN and serves as a stronger baseline than GAN under the first setting.

Fréchet Inception Distance (FID) (Heusel et al. (2017)) is used to evaluate the quality of generated images. FID calculates the Wasserstein-2 distance between generated images and real images (images with glasses) in the feature space of Inception-v3 network (Szegedy et al. (2015)). We train both networks for 600 epochs, and use WGAN-GP as the backbone for both GAN and DSGAN. In addition, transferring GANs are pretrained for 500 epochs, then being finetuned for 600 epochs.

Fig. 10 and Table 12 show generated images and FID for all methods, respectively. We can see that our DSGAN can generate images with glasses from the given $p_d$ and $p_{\bar{d}}$, and the FID of DSGAN are comparable to that of GAN. The experiment validates that DSGAN still works well to create complement data for complicate images.

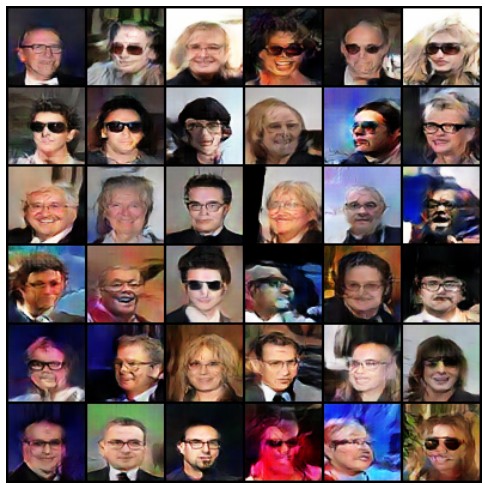

(a) Transferring GANs (pretrained with 20000 images and finetuned with 10000 samples with glasses)

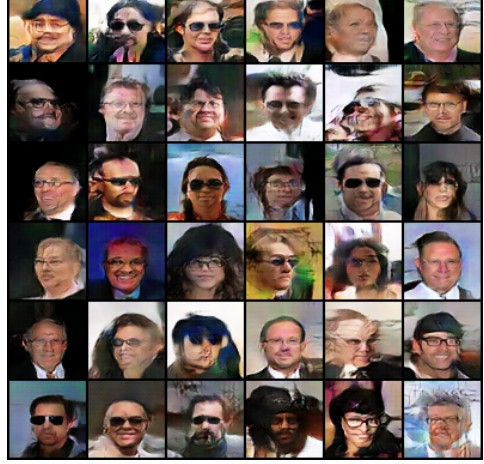

(b) DSGAN ($\alpha = 0.5$)

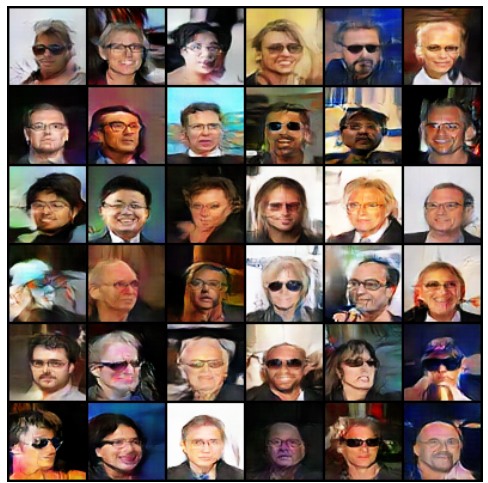

(c) Transferring GANs (pretrained with 50000 images and finetuned with 10000 samples with glasses)

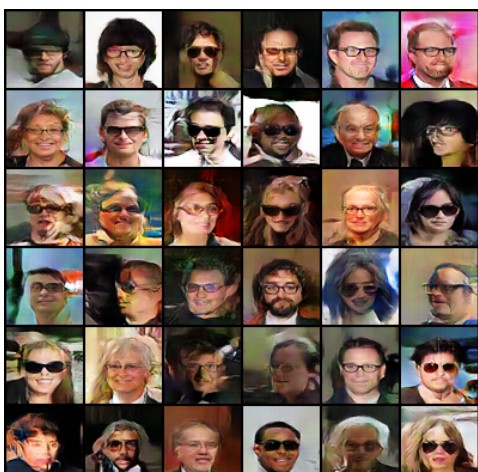

(d) DSGAN ($\alpha = 0.8$)

Figure 10: Sampled generated images of GAN and DSGAN on CelebA.

Table 12: FIDs of GAN and DSGAN on CelebA. Smaller FID means that the generated distribution is closer to the distribution of images with glasses.

|  | 10000 samples | 20000 samples | | 50000 samples | |
|---|---|---|---|---|---|
|  | GAN | transferring GAN | DSGAN ($\alpha = 0.5$) | transferring GAN | DSGAN ($\alpha = 0.8$) |
| FID | 22.37 | 18.34 | 18.05 | 16.45 | 15.39 |

