# OpenReview forum: "Difference-Seeking Generative Adversarial Network--Unseen Sample Generation"
_ICLR.cc/2020/Conference — Accept (Poster)_

### Official Review · AnonReviewer3 · 2019-10-23
**Official Blind Review #3**

**Rating:** 3

**Review:**

This paper proposed DSGAN which learns to generate unseen data from seen data distribution p_d and its somehow “broad” version p_{\hat d} (E.g., p_d convolved with Gaussian). The “unseen data” is the one that appears in p_{\hat d} but not in p_d. DSGAN is trained to generate such data. In particular, it uses samples from p_d as fake data and samples from p_{\hat d} as the real one.

Although the idea seems to be interesting, the paper seems to be a bit incremental and is a simple application of existing GAN techniques. The paper shows two applications (semi-supervised learning and novelty detection) and it is not clear that the proposed method outperforms existing GAN methods in the classification accuracy in MNIST/SVHN/CIFAR10 (Table 1) and existing sampling methods (Table. 3). It seems that the sampled reconstruction results (Fig. 8) are not as good as VAE on CIFAR10. I would also expect more ablation studies about how to pick p_{\had d}, which seems to be the key of this approach, in MNIST and CIFAR10.

In terms of writing, the paper is a bit confusing in terms of motivations and notations.

Overall, the method looks incremental and experimental results are mixed on small datasets so I vote for rejection. Note that I am not an expert on GAN/VAE so I put low confidence here.

**Experience Assessment:**

I have read many papers in this area.

**Review Assessment: Checking Correctness Of Derivations And Theory:**

I assessed the sensibility of the derivations and theory.

**Review Assessment: Checking Correctness Of Experiments:**

I assessed the sensibility of the experiments.

**Review Assessment: Thoroughness In Paper Reading:**

I read the paper at least twice and used my best judgement in assessing the paper.

---

> ### Author Response · Authors · 2019-11-10
> **Reply to review #3**
>
> Thanks for your comments! First, we have to clarify some misunderstandings.
>
>
> >>> it is not clear that the proposed method outperforms existing GAN methods in the classification accuracy in MNIST/SVHN/CIFAR10 (Table 1)
>
> BadGAN has already theoretically proved that complement data are helpful for semi-supervised learning. In this paper, we demonstrate that,  using our unseen data, the proofs in badGAN still can be satisfied but in a more concise way. Therefore, compared to badGAN that requires extra PixelCNN, DSGAN saves more computational memory and is time-efficienct.
>
>
> >>> It seems that the sampled reconstruction results (Fig. 8) are not as good as VAE on CIFAR10.
>
> In Novelty detection, we use the reconstruction error as a criterion to determine whether an image comes from seen class or unseen class. It is expected that images from the seen classes should be reconstructed better than those reconstructed from unseen classes. However, VAE cannot force the unseen classes with high reconstructed error. So, we combine DSGAN with VAE to deal with this issue. Due to the above reason, it is expected that "our sampled reconstruction results are not good as VAE". Note that the seen class, car, still can be reconstructed well by our method in Fig 8 (at the last row). The quantitative results in Table 3 further validate our approach.
>
>
> >>> I would also expect more ablation studies about how to pick p_{\had d}, which seems to be the key of this approach, in MNIST and CIFAR10.
>
> In fact, how to design $p_{\hat{d}}$ depends on applications  instead of datasets, as described in Sec. 4.1 and Sec. 4.2. Please note that, in Section 5.2.1, we used the same $p_{\bar{d}}$ for ALL datasets.
>
> We also want to clarify the datasets used in our experiments. In semi-supervised learning, we follow our competitors to conduct experiments on MNIST, SVHN and CIFAR10. In novelty detection, our method is evaluated on CIFAR10, which is also common in this application. Furthermore, we also add additional experiments about generating complement data in CelebA, which is a more complex dataset. We can see from Fig. 10 (Appendix G) that DSGAN can create complement data for complicate images well.

---

### Official Review · AnonReviewer2 · 2019-10-25
**Official Blind Review #2**

**Rating:** 6

**Review:**

Summary

This paper provides an interesting application of GAN which can generate the outlier distribution of training data which forces generator to learn the distribution of the low probability density area of given data. To show the effectiveness of the method, the author intuitively shows how it works on 2-D points data as well as the reconstructed Mnist dataset. Additionally, this approach reaches a comparable performance on semi-supervised learning and novelty detection task.

Paper Strengths

1. The idea of this paper is novel, and the implementation of this method is easily interacted with any GAN model. Also, due to its concise structure compared to the existing method, it saves more computational memory and is time efficiency.

Paper Weaknesses

1. Experimental settings are clear, however, what makes me confused is that the construction for p_{\bar{d}} is straightforward for simple distribution like 2D points dataset, however, it might be intractable for complex high dimensional data such as images.
2. The model seems to be sensitive to the hyper-parameter \alpha, is this parameter always fixed at 0.5 or needed to fine-tune for different datasets?

**Experience Assessment:**

I have published one or two papers in this area.

**Review Assessment: Checking Correctness Of Derivations And Theory:**

I assessed the sensibility of the derivations and theory.

**Review Assessment: Checking Correctness Of Experiments:**

I assessed the sensibility of the experiments.

**Review Assessment: Thoroughness In Paper Reading:**

I read the paper at least twice and used my best judgement in assessing the paper.

---

> ### Author Response · Authors · 2019-11-10
> **Reply for review #2**
>
> Thanks for your comments!
>
> >>> Experimental settings are clear, however, what makes me confused is that the construction for $p_{\bar{d}}$ is straightforward for simple distribution like 2D points dataset, however, it might be intractable for complex high dimensional data such as images.
>
> In responding to this comment and the comment of Reviewer #1, we perform one more experiment on CelebA to demonstrate that DSGAN can work well even for complicated images. In this experiment, we generate the color images of size 64 $\times$ 64. Similar to 1/7 experiments on the MNIST dataset, we let $p_{\bar{d}}$ be the distribution of face images with glasses and without glasses, and let $p_{d}$ be images without glasses.  We sample 10000 images with glasses and 10000 images without glasses from CelebA, and we set $\alpha$ to 0.5.
>
>
> In order to verify the generated image quality of DSGAN, we also train a GAN for comparison. GAN is trained with the same amount of training images (but only using face images with glasses since GAN is to learn the distribution of training data). In other words, we assume GAN can use complement data as training data directly. On the contrary, DSGAN only uses complement data indirectly (the difference between $p_{\bar{d}}$ and $p_d$).
>
> Figure 10 in Appendix G shows generated images and FID for both methods. We can see that our DSGAN can generate images with glasses from the given $p_d$ and $p_{\bar{d}}$, and the FID of DSGAN are comparable to that of GAN. The experiment validates that DSGAN still works well to create complement data for complicate images.
>
>
> >>> The model seems to be sensitive to the hyper-parameter $\alpha$, is this parameter always fixed at 0.5 or needed to fine-tune for different datasets?
>
> Since the optimal $\alpha$ of generating "unseen" data in DSGAN depends on the degree of overlap between $p_{\bar{d}}$ and $p_d$, it might need to be fine-tuned for different datasets. However, in our experiments, we set $\alpha$ to $0.8$ in most cases.
>
> Theorem 1 illustrates $\alpha$ should be expected to be as large as possible if both network G and D have infinite capacity. Though the networks never have the infinite capacity in real applications, a general rule is to pick a large $\alpha$ and force the complement data to be far from p_d, which is similar to the ablation studies in Sec. 5.1. According to our empirical observations, $\alpha = 0.8$ is the good choice for all datasets. Table 11 in Sec. F of Appendix shows the experimental results of how $\alpha$ affects the performances. We use different $\alpha$ values in the MNIST, SVHN and CIFAR10 dataset, respectively. One can see that we achieve the best performances at $\alpha = 0.8$.

---

### Official Review · AnonReviewer1 · 2019-11-06
**Official Blind Review #1**

**Rating:** 6

**Review:**

This paper proposed the DSGAN model to generate unseen data. The intuition based on standard GAN is straightforward and makes sense. The paper is well written, especially the case studies illustrate the idea clearly. The designing of p_{\bar{d}} also presented the limitation of this method. Two main discussed applications, semi-supervised learning and novelty detection are important in machine learning. In general, this is an interesting paper.

However, my concern is about the experiments. As a generative model for unseen data, I would like to see the generated results, which is more convincing. Only the 1/7 examples of MNIST dataset are provided in case studies. I am wondering for more complicated images, how is the performance?


**Experience Assessment:**

I have published one or two papers in this area.

**Review Assessment: Checking Correctness Of Derivations And Theory:**

I assessed the sensibility of the derivations and theory.

**Review Assessment: Checking Correctness Of Experiments:**

I assessed the sensibility of the experiments.

**Review Assessment: Thoroughness In Paper Reading:**

I made a quick assessment of this paper.

---

> ### Author Response · Authors · 2019-11-10
> **Reply for review #1**
>
> Thanks for your comments!
>
> >>> Only the 1/7 examples of MNIST dataset are provided in case studies. I am wondering for more complicated images, how is the performance?
>
> In responding to this comment and the first comment of Reviewer #2, we perform one more experiment on CelebA to demonstrate that DSGAN can work well even for complicated images. In this experiment, we generate the color images of size 64 $\times$ 64. Similar to 1/7 experiments on the MNIST dataset, we let $p_{\bar{d}}$ be the distribution of face images with glasses and without glasses, and let $p_{d}$ be images without glasses.  We sample 10000 images with glasses and 10000 images without glasses from CelebA, and we set $\alpha$ to 0.5.
>
> In order to verify the generated image quality of DSGAN, we also train a GAN for comparison. GAN is trained with the same amount of training images (but only using face images with glasses since GAN is to learn the distribution of training data). In other words, we assume GAN can use complement data as training data directly. On the contrary, DSGAN only uses complement data indirectly (the difference between $p_{\bar{d}}$ and $p_d$).
>
> Figure 10 in Appendix G shows generated images and FID for both methods. We can see that our DSGAN can generate images with glasses from the given $p_d$ and $p_{\bar{d}}$, and the FID of DSGAN are comparable to that of GAN. The experiment validates that DSGAN still works well to create complement data for complicate images.

---

### Author Response · Authors · 2019-11-10
**We upload the revised manuscript .**

Thanks for all the comments!

We list all the modifications as follows.

1. We add Sec. F in appendix for the ablation study on different $\alpha$.
2. We add Sec. G in appendix to demonstrate the sample quality of DSGAN on CelebA.

---

### Author Response · Authors · 2019-11-14
**We upload the revised manuscript .**

In this revision, our modifications are listed as follows.

1. We add more experiments and descriptions in Sec. G of Appendix because we trained a transferring GAN ([1]) as a strong competitor. However, compared with transferring GAN, our DSGAN still performs well. Please see Fig. 11 of Sec. G of Appendix for the new results.

[1] Wang, Yaxing et al. “Transferring GANs: generating images from limited data.” ECCV (2018).

---

### Decision · Program_Chairs · 2019-12-19

**Decision:**

Accept (Poster)

**Comment:**

The authors propose a way to generate unseen examples in GANs by learning the difference of two distributions for which we have access. The majority of reviewers agree on the originality and practicality of the idea.